# Unbiased Proteomic Exploration Suggests Overexpression of Complement Cascade Proteins in Plasma from Patients with Psoriasis Compared with Healthy Individuals

**DOI:** 10.3390/ijms25168791

**Published:** 2024-08-13

**Authors:** Bjørn Kromann, Lili Niu, Line B. P. Møller, Julie Sølberg, Karolina Sulek, Mette Gyldenløve, Beatrice Dyring-Andersen, Lone Skov, Marianne B. Løvendorf

**Affiliations:** 1Department of Dermatology and Allergy, Copenhagen University Hospital-Herlev and Gentofte, 2900 Hellerup, Denmark; 2Department of Dermatology, Zealand University Hospital, 4000 Roskilde, Denmark; 3Department of Clinical Medicine, Faculty of Health and Medical Sciences, University of Copenhagen, 2200 Copenhagen, Denmark; 4Novo Nordisk Foundation Center for Protein Research, Faculty of Health and Medical Sciences, University of Copenhagen, 2200 Copenhagen, Denmark; 5System Medicine, Steno Diabetes Center Copenhagen, 2730 Herlev, Denmark; 6Leo Foundation Skin Immunology Research Center, Faculty of Health and Medical Sciences, University of Copenhagen, 2200 Copenhagen, Denmark

**Keywords:** psoriasis, plasma, proteomics, complement system

## Abstract

Knowledge about the molecular mechanisms underlying the systemic inflammation observed in psoriasis remains incomplete. In this study, we applied mass spectrometry-based proteomics to compare the plasma protein levels between patients with psoriasis and healthy individuals, aiming to unveil potential systemically dysregulated proteins and pathways associated with the disease. Plasma samples from adult patients with moderate-to-severe psoriasis vulgaris (N = 59) and healthy age- and sex-matched individuals (N = 21) were analyzed using liquid chromatography–tandem mass spectrometry. Patients did not receive systemic anti-psoriatic treatment for four weeks before inclusion. A total of 776 protein groups were quantified. Of these, 691 were present in at least 60% of the samples, providing the basis for the downstream analysis. We identified 20 upregulated and 22 downregulated proteins in patients with psoriasis compared to controls (*p* < 0.05). Multiple proteins from the complement system were upregulated, including C2, C4b, C5, and C9, and pathway analysis revealed enrichment of proteins involved in complement activation and formation of the terminal complement complex. On the other end of the spectrum, periostin was the most downregulated protein in sera from patients with psoriasis. This comprehensive proteomic investigation revealed significantly elevated levels of complement cascade proteins in psoriatic plasma, which might contribute to increased systemic inflammation in patients with psoriasis.

## 1. Introduction

Psoriasis is a common systemic inflammatory disease with skin manifestations and is associated with several inflammatory comorbidities, including cardiometabolic disease [1]. Extensive research has revealed that T helper (Th) 17 cells and the interleukin (IL) 17/IL23 axis play a pivotal role in the pathogenesis of the disease [2]. However, knowledge of disease mechanisms in psoriasis remains incomplete, and investigations of circulating proteins are important to gain a comprehensive understanding of the systemic component of psoriasis and may provide the opportunity to discover new biomarkers and potential drug targets. Liquid chromatography–tandem mass spectrometry (LC-MS/MS)-based proteomics have the potential for such discoveries, given its untargeted nature and ability to identify hundreds of proteins in a single blood sample [3]. Hence, mass spectrometry and also panel-based targeted proteomics have already been applied in smaller studies comparing plasma or serum protein levels between healthy individuals and patients with psoriasis [4]. These studies identified several dysregulated proteins, particularly proteins related to IL-17 signaling and neutrophil activation or other innate immune responses. Two proteins from the complement system, complement C2 from the classical pathway and complement C5, which is a part of the terminal complement complex, were consistently observed to be increased in blood from patients with psoriasis across two and three studies, respectively, but only one study identified them both to be upregulated [5]. Complement proteins are known to be deposited locally in lesional psoriatic epidermis, but its role as a potential contributor to systemic inflammation in psoriasis is still unclear [6].

The aim of the current study was to improve existing knowledge about proteomic changes in psoriatic plasma and thereby identify potential biomarkers and drug targets. Therefore, we performed an in-depth proteomic comparison of plasma from patients with psoriasis and a matched group of healthy individuals using untargeted LC-MS/MS-based proteomics at a scale exceeding prior studies in the field.

## 2. Materials and Methods

Plasma from a previously described cohort of adult patients with psoriasis vulgaris and age- and sex-matched healthy individuals were included [7]. All participants were free from any autoimmune or dermatological diseases, other than psoriasis. The patients did not receive systemic anti-psoriatic treatment for at least four weeks prior to inclusion. For biologics and other drugs with longer half-lives, the minimal washout period was five half-lives. There were no restrictions regarding topical therapy. This study was approved by the Ethics Committee of the Capital Region in Denmark (H-19036270) and was conducted following the Declaration of Helsinki. All patients gave written informed consent before entry into the study.

Plasma samples were prepared according to a published protocol, and 500 ng of purified peptides was loaded to Evotips (Evosep Biosystems, Odense, Denmark) for the LC-MS/MS analysis [8,9]. All samples were analyzed on the EVOSEP One system using an in-house packed 15 cm, 150 µm diameter capillary column with 1.9 µm Reprosil-Pur C18 beads (Dr. Maisch, Tübingen, Germany) using a 21 min gradient with a flow rate of 1 µL/min. The column temperature was maintained at 60 °C using an integrated column oven (PRSO-V1, Sonation, Biberach, Germany) and interfaced online with the Orbitrap Exploris 480 MS with a field asymmetric ion mobility spectrometry device (FAIMS) at −45 V. For single-shot plasma data-independent analysis (DIA) experiments, 22 windows of dynamic sizes with an overlap of 1 Da were used, with full MS resolutions of 60,000 at *m*/*z* 200, and the full MS-normalized AGC target was set to 300% at a maximum injection time of 28 ms. MS/MS resolution was set to 30,000 at *m*/*z* 120, and the normalized AGC target of 10,000% was set at a maximum injection time of 28 ms, with 30% of HCD collision energy. Data were acquired in profile mode at 2650 V. DIA spectra were analyzed with Spectronaut (version 17, Biognosys, Zürich, Switzerland) with default settings for directDIA analysis mode with a false discovery rate (FDR) of 1% at peptide precursor and protein level. DIA data were matched against the human Uniprot database (January 2018).

Differential expression analysis of protein intensities was performed in R statistical software (version 4.3). We used the Wilcoxon rank sum test to compare healthy and psoriatic plasma protein levels with α = 0.05. Missing protein intensity measurements were imputed using Perseus (downshift 1.8 standard deviations, width 0.3 standard deviations) based on the log-transformed distribution of intensities of all protein measurements. STRING version 12.0 and Enrichr online-based tools were used for pathway analyses of significantly differentially abundant proteins with their inherent FDR control [10,11].

## 3. Results

We performed plasma proteome profiling on 59 patients with psoriasis and 21 healthy individuals (Table 1, Figure 1A).

In total, we quantified 776 protein groups. We filtered the data for proteins with quantitative values in more than 60% of the samples and included the remaining 691 protein groups in the downstream analyses (Appendix A). Principal component analysis (PCA) of all proteins did not separate patients with psoriasis and healthy individuals. The Wilcoxon rank sum test revealed 42 significantly differentially expressed proteins between the groups, including 20 proteins that were upregulated and 22 proteins that were downregulated in patients with psoriasis compared with healthy controls (*p* < 0.05) (Appendix A). Pathway analysis of upregulated proteins revealed significant enrichment of proteins related to complement activation and immune system pathways in patients with psoriasis compared with healthy individuals (FDR < 0.05) (Figure 1C). Among these proteins were complement factors C2, C4b, C5, and C9, which are all involved in the classical pathway of complement activation and terminal complement complex formation. In addition, SERPIN-G1, an inhibitor of spontaneous complement activation, and protein S100-A9, an important inflammatory protein in type 3 immunity that was earlier reported to be upregulated in psoriatic skin lesions and blood, were upregulated as well [4]. The soluble regulators of keratinocyte differentiation, dermokine and keratinocyte differentiation-associated protein, were upregulated, potentially indicating a spillover from the epidermis to the bloodstream. In contrast to the proteins that were increased in plasma from individuals with psoriasis, pathway enrichment analysis of the downregulated proteins did not yield significant pathway enrichment related to the immune system. We noted that periostin was the most downregulated protein in our analysis, with a 37% lower level in plasma from patients with psoriasis. Periostin is a marker of type 2 inflammation and has been shown to be downregulated in lesional psoriatic skin multiple times [4].

## 4. Discussion

We identified 776 proteins in an LC-MS/MS setup and based our analysis on the 691 proteins quantified in at least 60% of the samples. This study offers the most comprehensive proteomic comparison of plasma between patients with psoriasis and healthy individuals to date, in terms of both the number of included psoriasis patients and the total number of participants with deep proteomic coverage. Our data suggest that the overall proteomic composition of plasma is quite similar in patients with psoriasis compared with age- and sex-matched healthy individuals, although proteomic differences are present. Pathway analysis of the differentially expressed proteins points towards elevated levels of complement proteins in plasma from patients with psoriasis.

The fold changes in the expression of complement proteins in our study were relatively small and might thus have been overlooked in studies with lower statistical power. The complement system is a part of the innate immune system, and previous studies have shown involvement of the complement system in psoriasis, including pustular disease variants [6,12,13,14]. Recently it was reported that the serum levels of complement proteins before treatment were higher in responders to topical treatment, whereas the serum proteome in patients who did not respond to topical treatment had a more pronounced IL-17 signaling signature [15]. In line with this finding, the authors also observed the complement pathway overexpression to be attenuated after a successful topical treatment response which could indicate that the skin provides a source for the increased levels of complement proteins in circulation.

Even though the slight upregulation of complement-related proteins in plasma might not impose significant negative consequences on the individual patient, it is important to consider that it might affect the overall population of patients with psoriasis, e.g., in terms of inflammatory comorbidities such as cardiovascular disease [16]. The upregulated complement proteins C5 and C9 are implicated in the formation of the terminal complement complex, which has been shown to drive the cell death of macrophages and the proliferation of smooth muscle cells in atherosclerotic plaque formation. Furthermore, C2(b) and C4b are involved in the initiation and propagation of complement activation [17]. Mourino-Alvarez et al. recently identified alterations in complement proteins when comparing patients with psoriasis, with or without subclinical atherosclerosis. This suggests that a dysregulated complement system could contribute to the development of atherosclerosis in psoriasis [18]. In our study, SERPIN-G1, a serine protease inhibitor preventing spontaneous activation of the complement system, was also upregulated in patients with psoriasis potentially moderating the damaging effects of increased levels of circulating complement factors.

Periostin levels were decreased by 37% in plasma from patients with psoriasis in our study, which stands in contrast to a recent finding by Wojciechowska et al., who found periostin to be highly upregulated in serum from patients with psoriasis, although they at the same time showed a tendency towards a negative correlation between the Psoriasis Area and Severity Index (PASI) and periostin [19]. In a recent systematic review of proteomic studies in psoriasis, we identified four studies of different populations reporting periostin to be downregulated in psoriatic lesions, thus supporting that the downregulation rather than the upregulation of periostin might be a feature of psoriasis in most populations [4,20,21,22,23]. However, these studies have all measured protein levels in full-thickness skin punch biopsies and not in serum. Another recent study performed by Flink et al. provided further context regarding the role of periostin in psoriasis in multiple ways. First, they found that periostin was increased in keratinocytes in the epidermal basal layer and decreased in the dermis, which they concluded to be a result of basement membrane disruption and wound healing-like processes taking place in psoriatic lesions. Interestingly, they measured the highest periostin expression in healed lesions—not active lesions—thus indicating a potential beneficial effect of the protein. Furthermore, they also measured serum levels of periostin and, in contrast to our findings and in line with those of Wojciechowska et al., they also report serum periostin to be increased in psoriasis. Interestingly, they see that this is driven by systemically treated patients and not untreated patients, which would explain why we did not replicate this finding [24]. These somewhat conflicting findings and the fact that periostin seems to be elevated in the sera of systemically treated patients, in patients with lower PASI, and healed lesions make an intriguing call to further examine the role of periostin in psoriasis. If periostin improves rather than aggravates the inflammation in psoriasis, this might also turn out to be true in other inflammatory diseases as well, such as rheumatoid arthritis, where periostin was elevated in patients with disease remission compared with healthy controls [25]. Lastly, periostin has recently been suggested to be a marker to discriminate atopic dermatitis from psoriasis since it is known to be very highly expressed in atopic dermatitis, where it positively correlates with disease activity [26].

The limitations of our study include the use of nominal *p*-values in the differential expression analysis, limited sample size, and the observational and cross-sectional study design. Additionally, it is important to note that mass spectrometry has its limitations, particularly in the presence of highly abundant proteins like albumin, which hinders the detection of proteins with very low abundance. We chose not to deplete albumin from the samples in order to prevent the simultaneous removal of proteins bound to albumin, and despite this strategy, we quantified a substantial number of proteins in this hypothesis-generating study.

In conclusion, our comparative analysis of the 691 different plasma proteins indicates an upregulation of the complement cascade pathway in patients with psoriasis compared with healthy individuals. Functional investigations of the activity in the complement system could increase knowledge about the role of this part of the innate immune system in psoriasis and its comorbidities. Among the downregulated proteins identified in our analysis, periostin stands out as a marker with a possible relation to a clinical improvement in psoriasis, although more research is warranted to examine this further.

## Figures and Tables

**Figure 1 ijms-25-08791-f001:**
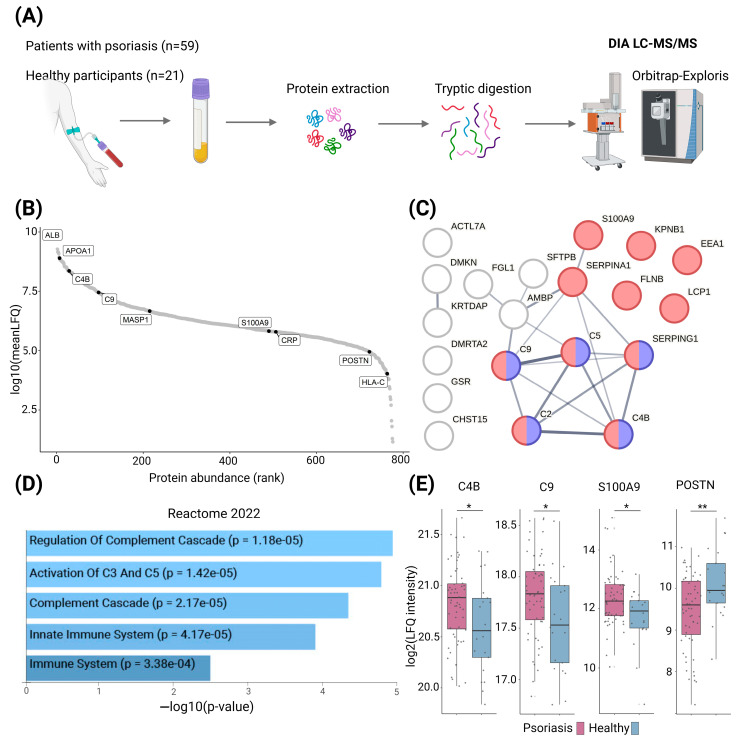
Plasma-derived proteins from patients with psoriasis reveal enrichment of the complement cascade. (**A**) In this study, we analyzed the proteomic differences between plasma from 59 patients with psoriasis and 21 healthy individuals by DIA LC-MS/MS. (**B**) Rank plot showing the overall mean label-free quantification (LFQ) intensities of the quantified proteins. As expected, albumin was the most abundant protein in this study of non-depleted plasma. (**C**) String network analysis of the 20 upregulated proteins (*p* < 0.05). The red nodes are proteins present in the ‘immune system’. Reactome pathway and blue nodes represent proteins from the ‘Complement activation, classical pathway’ in the Gene Ontology knowledgebase. The string interaction score cut-off was set to 0.4. (**D**) Enrichment analysis based on the 20 upregulated proteins. The bar chart shows the top five enriched terms from the Reactome database, along with their corresponding *p*-values. All *p*-values remained significant after correction for multiple testing (FDR < 0.05). (**E**) Boxplots of protein intensities of selected proteins in plasma from patients with psoriasis compared with healthy controls (* = *p* < 0.05; ** = *p* < 0.01). Abbreviations: DIA = data-independent acquisition; LC-MS/MS = liquid chromatography–tandem mass spectrometry; LFQ = label-free quantification.

**Table 1 ijms-25-08791-t001:** **Participant characteristics**.

	Psoriasis (N = 59)	Healthy (N = 21)
**Male sex**	40 (67.8%)	16 (76.2%)
**Age, mean (SD)**	45.0 y (16.8)	41.6 y (15.7)
**PASI, mean (SD)**	11.9 (5.1)	NA
**Disease duration, mean (SD)**	20.7 y (14.5)	NA
**Receiving topical treatment at inclusion**	25 (42.4%)	NA
**Psoriatic arthritis**	8 (13.6%)	0 (0%)

Abbreviations: SD = standard deviation, NA = not applicable, PASI = Psoriasis Area and Severity Index, y = years.

## Data Availability

Curated data are available in Appendix A. Raw data will be uploaded to the PRIDE repository and released upon publication.

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
