# Peer review of "Unbiased Proteomic Exploration Suggests Overexpression of Complement Cascade Proteins in Plasma from Patients with Psoriasis Compared with Healthy Individuals"

_ijms, 2024, doi:10.3390/ijms25168791_

Round 1

Reviewer 1 Report

Comments and Suggestions for Authors

Dear autors,

I read your manuscript with great ineterest. Proteomic is an elegant method to identify new thrapeutic, prognostic and diagnostic markers in inflammatory skin diseases including psoriasis.

The manuscript is well-written and easy to read.

My only comment is as below:

It would be interesting to have a table on demography of psoriasis patients included in this study especially with information on which treatment they are receiving or have received. It is not clear for me whether psoriasis patients included in this study were on topical/systemic treatments and the treatment was stopped for 4 weeks prior serum collection or they are naive to systemic or topical therapies. In case these patients have been on systemic treatments, further information is very necessary on which type of treatment. As we know the third generation biologics targeting IL-23 are applied every 8-12 weeks. And even if samples are collected between drug application intervals these patients are "under treatment" and this could influence the proteomic analysis and the result.

I find the manuscript suitable for publication after minor revision answering my above mentioned concern.

Author Response

Dear reviewer,

First of all we would like to thank you for taking your time to read through and provide your feedback on our work. It is highly appreciated and we find that the changes you have proposed are valid and thoughtful.

Comment 1: "It would be interesting to have a table on demography of psoriasis patients included in this study especially with information on which treatment they are receiving or have received. It is not clear for me whether psoriasis patients included in this study were on topical/systemic treatments and the treatment was stopped for 4 weeks prior serum collection or they are naive to systemic or topical therapies. In case these patients have been on systemic treatments, further information is very necessary on which type of treatment. As we know the third generation biologics targeting IL-23 are applied every 8-12 weeks. And even if samples are collected between drug application intervals these patients are "under treatment" and this could influence the proteomic analysis and the result."

Response 1: Thank you for pointing out that we could improve the clarity of our inclusion criteria and expanding Table 1 with more demographical information. We have revised the wording of the inclusion criteria under "2. Materials and Methods" on page 2, lines 72-73. We hope it is clear now that the patients were not undergoing active systemic treatment and that the wash-out period of at least five half-lives was sufficient.

Furthermore, we have expanded Table 1 under "3. Results" on page 3, line 128 with additional demographical information including information about what fraction of the patients received topical treatment at inclusion but also information about psoriatic arthritis.

Reviewer 2 Report

Comments and Suggestions for Authors

Your efforts are appreciated.

On what base you mentioned in the title that this study is unbiased, what are the precautions you took to prevent any bias in the study?

Introduction

Psoriasis now is considered as a systemic disease with skin manifestations not a skin disease with associated manifestations.

Aim

Add to aim( this is study was conducted to discover new biomarkers and drug targets).

Methods

The inclusion criteria of the patients must be rewritten.

I think 4 weeks we're not enough period to eliminate effects of systemic treatments of psoriasis from blood as some drugs like acitretin may take very long time to disappear from patient plasma so it was better to take de Novo patients who did not receive any treatment before as a patient group.

I think also topical treatment can ameliorate the results, so it is better avoided.

You must mention that the patients were free from other dermatological diseases, any inflammatory diseases, or any immunological diseases because if any was present a bias would appear in the results.

Limitations

In the limitations you must mention that small numbers of the patients end controls were also a limitation in this study.

Author Response

Dear reviewer,

First of all we would like to thank you for taking your time to read through and provide your feedback on our work. It is highly appreciated and we find that the changes you have proposed are valid and thoughtful.

Comment 1: "On what base you mentioned in the title that this study is unbiased, what are the precautions you took to prevent any bias in the study?"

Response 1: Thank you for asking this highly relevant question. Here, we used the word 'unbiased' because it is commonly used in explorative proteomic studies based on mass spectrometry. It refers to the fact that we did not select the proteins to measure beforehand which is where the method stands out from other proteomic methods such as Olink, SomaScan etc. which are panel-based and thus only provide the opportunity to identify proteins that are included in the selected panel(s).

Comment 2: Introduction: "Psoriasis now is considered as a systemic disease with skin manifestations not a skin disease with associated manifestations."

Response 2: Thank you for highlighting this important fact. We have revised the text accordingly in "1. Introduction" on page 1, lines 40-41.

Comment 3: Aim: "Add to aim( this is study was conducted to discover new biomarkers and drug targets)."

Response 3: Thank you for helping us introducing a clearer aim to our study. We have revised the wording accordingly in "1. Introduction" on page 2, lines 61-63.

Comment 4: Methods: "The inclusion criteria of the patients must be rewritten. I think 4 weeks we're not enough period to eliminate effects of systemic treatments of psoriasis from blood as some drugs like acitretin may take very long time to disappear from patient plasma so it was better to take de Novo patients who did not receive any treatment before as a patient group."

Response 4: Thank you for proposing this highly relevant concern regarding systemic treatment. We have used the criteria of 4 weeks or more without treatment because this is commonly used and accepted in clinical trials as long as the drugs do not have long half lives. The only systemic drug that some of the included patients had received before were methothrexate (e.g. no biologics, acitretin et.c), which is why we felt confident with a four-week washout period before the blood sample for this study was drawn. In the manuscript, we have clarified this in "2. Materials and Methods" on page 2, lines 71-73. Still we acknowledge your point that de novo patients with no prior treatment would have been optimal but such strict criteria would have lead to a diminished sample size which is why we included some patients with prior treatment.

Comment 5: Methods: "I think also topical treatment can ameliorate the results, so it is better avoided."

Response 5: Thank you for highlighting the concern that topical treatment might affect blood proteomics. While this might indeed be true we included these patients as they we judged them to be clearly undertreated based on their clinical appearance. This is underlined by the realtively high mean PASI of 11.9. Furthermore, we know that systemic uptake of topical treatments can happen but when treatment instructions are followed systemic levels of the topically applied drugs will be very low. Therefore we chose to include these patients for this study although we will take your point into consideration when planning future studies.

Comment 6: Methods: "You must mention that the patients were free from other dermatological diseases, any inflammatory diseases, or any immunological diseases because if any was present a bias would appear in the results."

Response 6: Thank you for pointing out that we should describe our inclusion/exclusion criteria more in detail. We have elaborated accordingly in the manuscript in "2. Materials and Methods" on page 2, lines 69-70.

Comment 7: Limitations: "In the limitations you must mention that small numbers of the patients end controls were also a limitation in this study."

Response 7: Thank you for your comment. We completely agree that this is still a limitation even though our sample size was bigger than previous studies in the field. We have added the suggested limitiation in "4. Discussion" on page 6, line 212.

Round 2

Reviewer 2 Report

Comments and Suggestions for Authors

your efforts are appreciated. You mentioned that your patients were on methotrexate therapy. So you have to mention this in the methodology.

Sector